**PLOS . Biology**

DISCOVERY REPORT

# An animal toxin-antidote system kills cells by creating a novel cation channel

Lews Caro[1,2], Aguan D. Wei[3], Christopher A. Thomas[4], Galen Posch[2], Ahmet Uremis[2], Michaela C. Franzi[4], Sarah J. Abell[4], Andrew H. Laszlo[4], Jens H. Gundlach[4], Jan-Marino Ramirez[3,5], Michael Ailion[1,2]*

**1** Molecular and Cellular Biology Ph.D. Program, University of Washington, Seattle, Washington, United States of America, **2** Department of Biochemistry, University of Washington, Seattle, Washington, United States of America, **3** Norcliffe Foundation Center for Integrative Brain Research, Seattle Children's Research Institute, Seattle, Washington, United States of America, **4** Department of Physics, University of Washington, Seattle, Washington, United States of America, **5** Department of Neurological Surgery, University of Washington School of Medicine, Seattle, Washington, United States of America

* mailion@uw.edu

## Abstract

Toxin-antidote systems are selfish genetic elements composed of a linked toxin and antidote. The *peel-1 zeel-1* toxin-antidote system in *C. elegans* consists of a transmembrane toxin protein PEEL-1 which acts cell autonomously to kill cells. Here we investigate the molecular mechanism of PEEL-1 toxicity. We find that PEEL-1 requires a small membrane protein, PMPL-1, for toxicity. Together, PEEL-1 and PMPL-1 are sufficient for toxicity in a heterologous system, HEK293T cells, and cause cell swelling and increased cell permeability to monovalent cations. Using purified proteins, we show that PEEL-1 and PMPL-1 allow ion flux through lipid bilayers and generate currents which resemble ion channel gating. Our work suggests that PEEL-1 kills cells by co-opting PMPL-1 and creating a cation channel.

## Introduction

Selfish genetic elements ensure their inheritance, even at the expense of host fitness. Toxin-antidote (TA) systems are one class of selfish element, made up of genetically linked toxin and antidote genes. The toxin is cytoplasmically inherited across generations while the cognate antidote is expressed in progeny. In animal TA systems, the parental toxin is transmitted to progeny via the sperm or egg. Offspring that do not inherit the TA genetic element are affected by the toxin, typically resulting in death or developmental defects [1], whereas the offspring which inherit the TA genetic element express the cognate antidote to prevent toxicity. Therefore, TAs guarantee their presence in the next generation by killing non-inheriting offspring. Recent work suggests that TA elements are more common among animals than previously thought [2,3]. However, the mechanisms of toxin and antidote activity remain largely unknown for animal TA systems.

**PLOS Biology**

**Data availability statement:** All relevant data are within the paper and its Supporting Information files.

**Funding:** This work was supported by National Science Foundation grants MCB-1552101 and MCB-2344838 to MA; HHS National Institutes of Health (NIH) T32 GM007270 to LC; NIH R01 HL144801, R01 HL151389, and R01 HL126523 to J-MR; and NIH R01 HG005115 to AHL and JHG. The funders had no role in study design, data collection and analysis, decision to publish, or preparation of the manuscript.

**Competing interests:** The authors have declared that no competing interests exist.

**Abbreviations :** AH, amphipathic helix; ER, endoplasmic reticulum; LDH, lactate dehydrogenase; MBP, maltose-binding protein; PM, plasma membrane; PMP3, plasma membrane proteolipid 3; TA, toxin-antidote; VRAC, volume-regulated anion channel.

One of the best characterized animal TA systems is *peel-1*/*zeel-1* in *C. elegans* [4]. The *peel-1* toxin is expressed during sperm development but is not toxic to sperm. Mature sperm carry PEEL-1 protein and fertilization delivers the toxin to embryos, resulting in developmental arrest [5]. However, offspring which inherit the TA element express the antidote *zeel-1* and do not arrest (Fig 1A). Expression of PEEL-1 in adult worms also causes death [5], suggesting that toxicity is not specific to a particular developmental stage. Furthermore, PEEL-1 acts cell-autonomously in *C. elegans*; ectopic expression of PEEL-1 in specific tissues causes death of those cells, with no defects in neighboring cells [5]. So far, no adult somatic cells have been found to be immune to PEEL-1, suggesting that toxicity works by disrupting a fundamental cellular process. In this study, we dissect the molecular mechanism of PEEL-1 toxicity. We find that PEEL-1 co-opts a conserved membrane protein of unknown function, PMPL-1. Together these proteins are sufficient to create a toxic, cation leak channel. Thus, our work determines the molecular mechanism of toxicity of an animal toxin-antidote system.

## Results

### PMPL-1 is required for PEEL-1 toxicity

To identify other genes required for PEEL-1 toxicity, we performed a large forward genetic screen for PEEL-1 suppressors in *C. elegans*. We mutagenized worms carrying transgenes for heat-shock inducible *peel-1* expression (*hsp-16.41p*::*peel-1*) [5]. We isolated only two full-suppressors of heat-shock PEEL-1 toxicity, and both suppressors carry mutations in F47B7.1 (hereafter named *pmpl-1*) (Fig 1B). Killing by endogenous, sperm-delivered PEEL-1 is also suppressed by *pmpl-1* mutations (Fig 1C), but not via a paternal-effect (S1 Fig), suggesting that PMPL-1 acts in embryos to facilitate PEEL-1 toxicity and does not act in sperm. *pmpl-1* codes for a 59 amino acid protein predicted to be an integral membrane protein (Fig 1D). The *pmpl-1* mutants were identified as a missense allele (*yak52*, A47T) and a full-gene deletion allele (*yak103*) (see "Materials and methods").

The *pmpl-1* expression pattern is consistent with its role in PEEL-1 toxicity. The *pmpl-1* promoter drives expression in embryos prior to toxicity from sperm-delivered PEEL-1 (S2A Fig) [5]. *pmpl-1* is also widely expressed in several adult tissues (S2B Fig), consistent with heat-shock PEEL-1 toxicity in adults [5]. Publicly available RNA-seq data [6] indicate that *pmpl-1* expression levels are lowest in the male gonad compared to all other tissues, consistent with wild-type sperm being unaffected by PEEL-1 (S2C Fig). These data suggest that PMPL-1 is required for cell susceptibility to PEEL-1 toxicity. We further confirmed that co-expression of PEEL-1 and PMPL-1 in the vulval muscle cells of *pmpl-1* mutants caused specific toxicity in this tissue (Figs 1E and S2D). Defects were not observed in surrounding tissues, indicating that PEEL-1 and PMPL-1 act in the same cell to cause toxicity.

### PMPL-1 is a conserved membrane protein of unknown function

PMPL-1 belongs to the Plasma Membrane Proteolipid 3 (PMP3) family of proteins, so we named it "PMP3-Like protein 1." PMP3 proteins are widely present in bacteria,

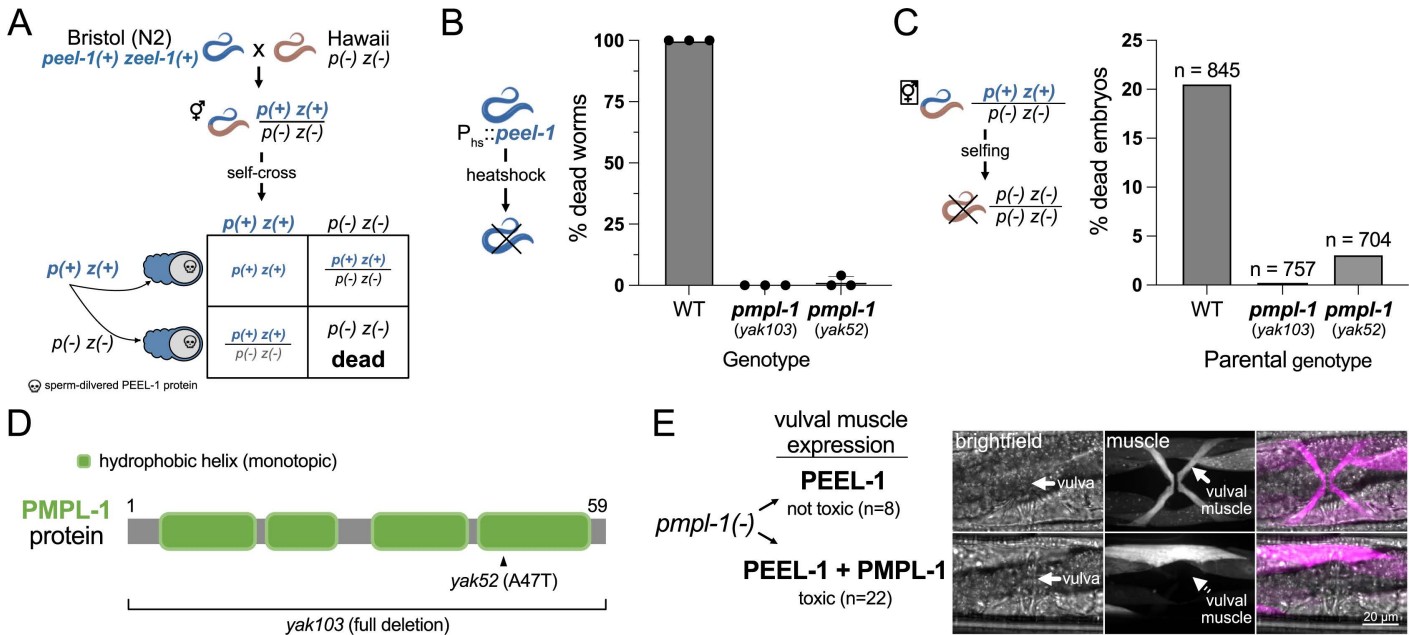

**Fig 1. PMPL-1 is necessary for PEEL-1 toxicity in *C. elegans*. (A)** Selfish activity of the *peel-1 zeel-1* toxin-antidote system in *C. elegans* shown in a genetic cross of strains from Bristol (which has the genetic element) and Hawaii (which lacks the genetic element). Hermaphrodite worms heterozygous for the presence of *peel-1 zeel-1* (*p(+) z(+)/p(−) z(−)*) have 25% inviable progeny. This is due to sperm-delivered PEEL-1 toxicity causing developmental arrest of *zeel-1(−)* progeny. **(B)** Proportion of worms dying from ectopic, heat shock-PEEL-1 expression. Two *pmpl-1* mutant alleles (*yak103* and *yak52*) provide resistance to toxicity. **(C)** Proportion of dead, arrested embryos from self-fertilizing hermaphrodites heterozygous for *peel-1 zeel-1*. n = total progeny scored. **(D)** Predicted domain structure of the PMPL-1 protein with mutant alleles shown. The hydrophobic helices are predicted to be monotopic, passing through one leaflet of a lipid bilayer. **(E)** Body wall and vulval muscle (magenta) of *pmpl-1(yak103)* worms with vulval muscle-specific expression of PEEL-1 alone (top) or PEEL-1 and PMPL-1 (bottom). Vulval muscles appears normal with PEEL-1 alone but are missing or atrophied when PEEL-1 and PMPL-1 are co-expressed in these cells. All worms expressing PEEL-1 and PMPL-1 in vulval muscle cells had missing or severely deformed vulval muscles (n = 22). *peel-1* and *pmpl-1* are both GFP tagged. All channels are shown in S2D Fig. Scale bar = 20 μm. Underlying data are available in S1 Data.

plants, and fungi, and are found in some simple animals [7]. The role of PMP3 proteins in animals is unknown, but PMP3 proteins in plants, fungi, and bacteria are important for cold-stress resistance, membrane protein trafficking, and ion homeostasis [8–11]. *pmpl-1* mutant worms do not have any obvious phenotypes. However, PMPL-1 is highly conserved among nematodes (S3 Fig). We identified 15 PMP3-like proteins in *C. elegans* (S4 Fig) through BLAST searches. PMPL-2 (also known as TXT-9) is most similar to PMPL-1 (75% similarity) and was found in an RNAi screen for defective transcellular chaperone signaling [12]. However, there is no known molecular function for any *C. elegans* PMP3-like protein.

PMP3 proteins are thought to contain two transmembrane spanning domains [13,14], consistent with DeepTMHMM predictions for PMPL-1 (S5A Fig) [15]. However, AlphaFold2 predicts PMPL-1 as a monotopic protein, with four helices passing through only one leaflet of a lipid bilayer (Figs 1D and S5B) [16,17]. We favor the monotopic prediction of PMPL-1 because two recently solved structures of a bacterial photosynthetic complex showed a PMP3 protein within the complex having a similar monotopic structure [18,19].

In contrast to *pmpl-1*'s broad conservation in nematodes (S3 Fig), *peel-1* is found only in *C. elegans* and has no homology to any known protein [5]. Additionally, *pmpl-1* is genetically unlinked from *peel-1 zeel-1*. These features suggests that PMPL-1 has biological roles other than supporting PEEL-1 toxicity and that PEEL-1 co-opts PMPL-1 for its own use.

## PEEL-1 and PMPL-1 are sufficient for toxicity in HEK293T cells

Given that *pmpl-1* was the only full suppressor of PEEL-1 toxicity found in our screen, we hypothesized that PEEL-1 and PMPL-1 may be the only two components required for toxicity. We expressed these proteins in human embryonic kidney cells (HEK293T) and assayed for cytotoxicity using lactate dehydrogenase (LDH) release into the culture media as a measure of plasma membrane rupture [20]. We found that each protein alone was not toxic, but co-expression of PEEL-1 and PMPL-1 resulted in significant cytotoxicity (Fig 2A). Other PMP3-like proteins such as *C. elegans* PMPL-2 and yeast PMP3 were not toxic with PEEL-1 (Fig 2B), suggesting that PMPL-1 has a specific role in toxicity that is not universal

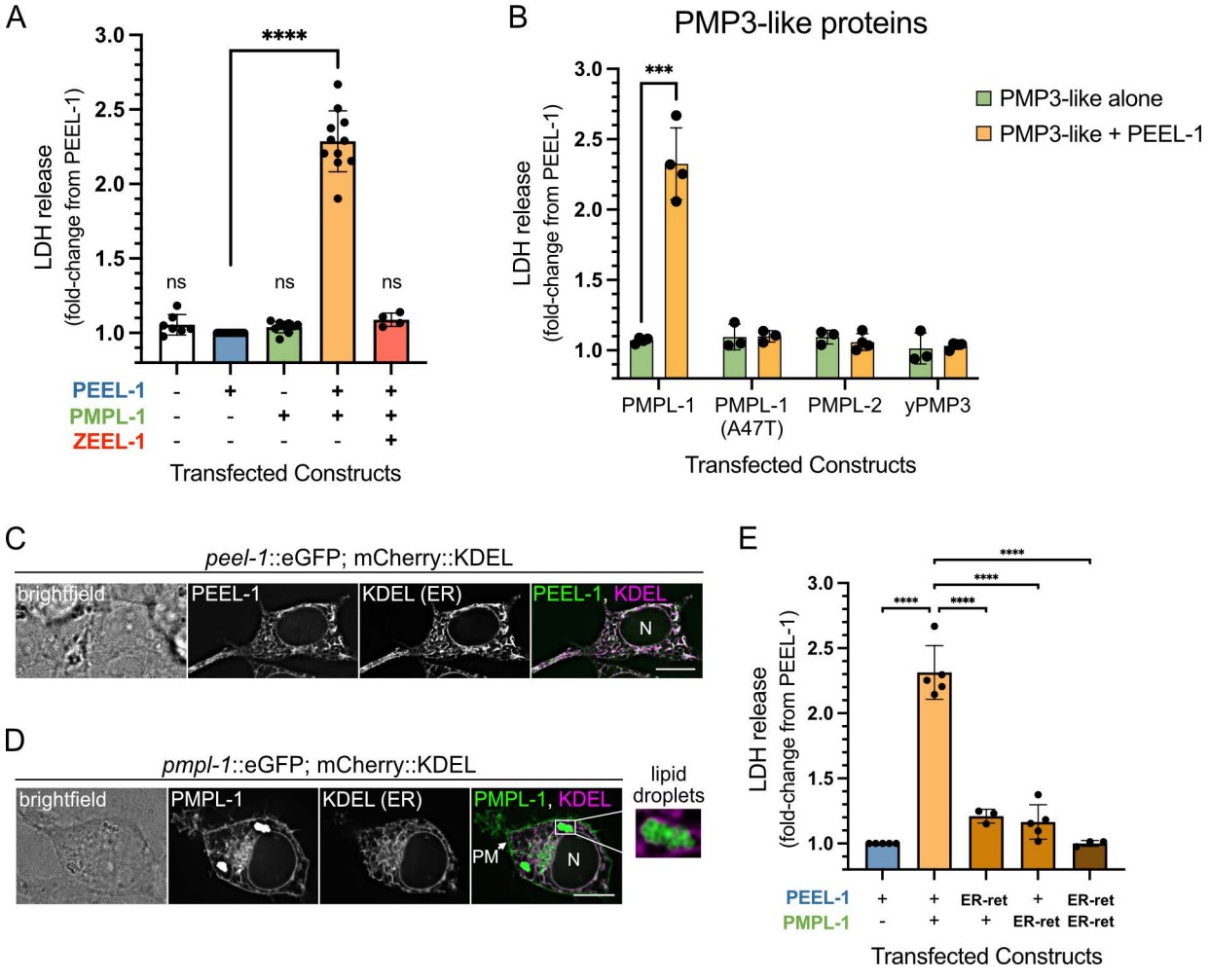

**Fig 2. PEEL-1 and PMPL-1 are sufficient for toxicity in HEK293T cells. (A)** Cytotoxicity (measured by LDH release) of combinations of three constructs transfected in HEK293T cells. Each data point is a biological replicate, normalized to LDH release from transfection with *peel-1*::eGFP in the same experiment. All plots show means with SD. Transfections combine constructs coding for mCherry or eGFP (−) or a fluorescent-tagged protein (+): PEEL-1::eGFP (top), PMPL-1::mCherry (middle), or mCherry::ZEEL-1 (bottom). **(B)** Cytotoxicity of PMP3-like proteins alone or with PEEL-1::eGFP. The PMPL-1 *yak52* (A47T) mutant protein, *C. elegans* PMPL-2, and the yeast homolog yPMP3 are shown. **(C)** Live-cell imaging of a single cell transfected with an ER-marker (mCherry::KDEL) and *peel-1*::eGFP or **(D)** *pmpl-1*::eGFP. The cell nucleus is indicated (N). PMPL-1 is also seen on the plasma membrane (PM) and on lipid droplets (inset). Scale bar = 10 μm. **(E)** Cytotoxicity is suppressed by addition of the GBR1 ER-retention tag on the C-terminus of PEEL-1::eGFP or PMPL-1::mCherry. *P*-values in (A) and (E) calculated using one-way ANOVA with Dunnett's multiple comparisons test, comparing all samples to PEEL-1 alone in (A) and all samples to PEEL-1 with PMPL-1 in (E). In (B), multiple unpaired *t*-tests were used with Holm–Šídák test, comparing each PMP3-like protein alone to PMP3-like with PEEL-1 (***$p < 0.001$; ****$p < 0.0001$). Underlying data are available in S1 Data.

to the PMP3 family. The reconstitution of toxin activity in a heterologous system suggests that PEEL-1 and PMPL-1 are sufficient for toxicity and may act by disrupting an essential, conserved cellular process. Antidote activity could also be reconstituted in HEK293T cells by co-expression of ZEEL-1, resulting in reduced toxicity (Fig 2A).

## Plasma membrane localization of PEEL-1 and PMPL-1 is critical for toxicity

PEEL-1 and PMPL-1 localize to several membrane-bound compartments in HEK293T cells and *C. elegans*. Both proteins localize to the endoplasmic reticulum (ER), nuclear envelope-associated ER, and plasma membrane (PM) (Figs 2C–2D and S6B–6C). Additionally, PMPL-1 localizes to lipid droplets (Figs 2D and S6A), consistent with the predicted monotopic topology. Although PEEL-1::eGFP is not easily detectable on the PM of transfected HEK293T cells, stable expression of PEEL-1 in HEK293 cells or in *pmpl-1* knock-out worms shows clear PM localization (S6B–6C Fig). This also suggests that PEEL-1's trafficking to the plasma membrane does not require PMPL-1. Since PEEL-1 and PMPL-1 co-localize on the ER and PM, we hypothesized that they act together at one of these locations to induce toxicity.

To determine in which compartment these proteins perform their toxic roles, we prevented their movement to the PM by the addition of a GBR1 ER-retention tag [21]. ER-retention tags on either protein resulted in more than an 80% drop in toxicity, and retention tags on both proteins completely suppressed toxicity (Fig 2E), suggesting that PEEL-1 and PMPL-1 do not act in the ER and likely perform their toxic roles at the PM.

## PEEL-1 toxicity causes cell swelling in HEK293T cells

LDH release is an end-point measure of plasma membrane rupture, but it was unclear whether plasma membrane rupture was a primary or secondary effect of toxicity. Imaging 48 hrs after transfection showed that 92% of HEK293T cells co-expressing PEEL-1 and PMPL-1 were swollen, about double their typical size (Fig 3A–3B). Live imaging showed cells exhibiting abnormal phenotypes approximately 16 hrs after transfection. Cells began with slight swelling of the nucleus and cell, followed by jetting out round protrusions of 5−10 μm, similar in size to the nucleus (Fig 3C and S1 Movie). These large protrusions are short-lived, typically existing for less than 10 min before being reabsorbed by the cell. Eventually, cells swell evenly in all directions (Fig 3D). Swollen cells often lose the integrity of their plasma membrane, followed by ER fragmentation and ER swelling (S7 Fig). By 48 hrs after transfection, we observe various outcomes for swollen cells: cell lysis, cell detachment from the plate, and swollen, intact cells attached to the plate. Our LDH assay likely captures only the first of these phenotypes and may underestimate the fraction of cells experiencing toxicity. The cell swelling phenotype is reminiscent of the necrotic vacuoles previously described in PEEL-1 affected embryos in *C. elegans* [5].

The cell swelling phenotype caused by PEEL-1 toxicity suggests that cell death is non-apoptotic [22]. Indeed, *C. elegans* mutants defective in apoptosis (*ced-3*) and apoptotic cell engulfment (*ced-2* and *ced-5*) are still susceptible to PEEL-1 toxicity (S8 Fig) [23–25]. Other mammalian cell death pathways such as pyroptosis and necroptosis appear to be absent from *C. elegans* [26]. Thus, the cytotoxic phenotypes we observe are likely due to a direct effect of PEEL-1 and PMPL-1 activity rather than indirect induction of programmed cell death.

## PEEL-1 has an amphipathic helix that is critical for toxicity

PEEL-1 has no homology to known proteins [5], so we turned to structural predictions to identify important regions in PEEL-1. AlphaFold2 predicted a low-confidence structure with six alpha-helices (Fig 4A) [16,17]. The longest four helices matched DeepTMHMM's prediction of four transmembrane domains (S5C Fig). Closer inspection revealed that the fourth predicted transmembrane domain is amphipathic (Fig 4B). An amphipathic helix (AH) has opposing hydrophilic and hydrophobic faces and is a critical feature of known pore-forming toxins. For example, actinoporins are a diverse family of toxins which oligomerize their AHs to construct channels [27]. The properties of the predicted PEEL-1 AH are similar to actinoporin AHs. The PEEL-1 AH is 22 amino acids in length (residues 111−132), making it long enough to span the lipid bilayer [28]. The hydrophobic moment (μH), a measure of the degree of amphipathicity, is within the range of actinoporin

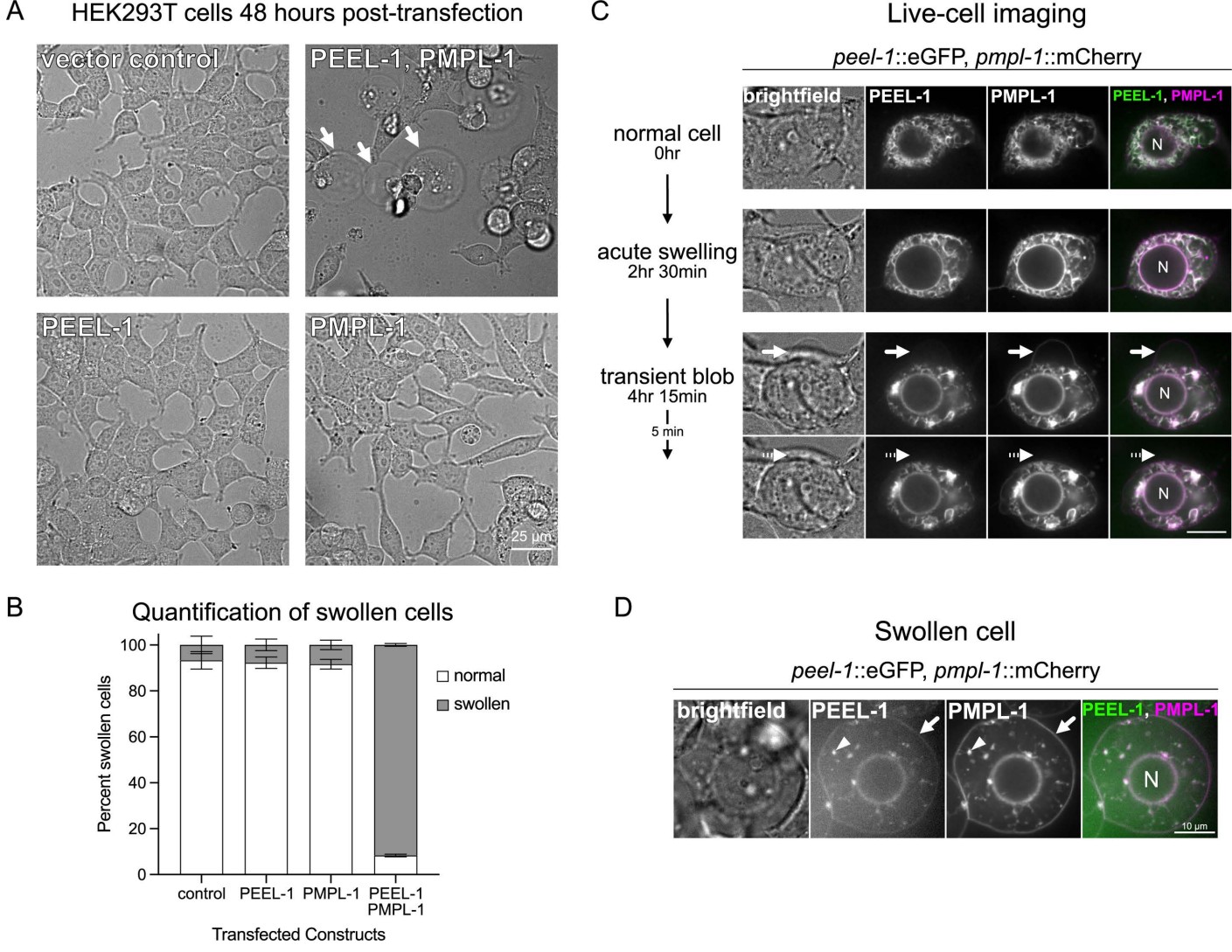

**Fig 3. PEEL-1 toxicity results in HEK293T cell swelling. (A)** Brightfield images of cells transfected with indicated constructs. Cells were imaged 48 hrs after transfection. Scale bar = 25 μm. Arrows point to swollen cells. **(B)** Percent of transfected cells that appear normal or swollen under brightfield. Mean with SD of three biological replicates is shown. One hundred cells were scored for each condition in each biological replicate. **(C)** Selected frames of a live-cell imaging time-course experiment (see S1 Movie). A single cell is shown, co-expressing PEEL-1::eGFP and PMPL-1::mCherry. The nucleus is labeled ("N"). $t = 0$ is 16 hrs after transfection. Acute swelling can be seen ($t = 2$ hr 30 min), followed by a transient blob or protrusion (arrow) jetted out by the cell ($t = 4$ hr 15 min) and later reabsorbed (dotted arrow) ($t = 4$ hr 20 min). Scale bar = 10 μm. **(D)** A typical phenotype from a cell transfected with PEEL-1::eGFP and PMPL-1::mCherry at 48 hrs post-transfection. PEEL-1 and PMPL-1 can be seen on the plasma membrane (arrow) and in fragmented ER (arrowhead). Scale bar = 10 μm. Underlying data are available in S1 Data.

channel-forming toxins (S9 Fig) [29]. One notable difference is that the PEEL-1 AH is more hydrophobic than actinoporin AHs (S9 Fig), suggesting that the PEEL-1 AH may always exist within the membrane, unlike in actinoporins which have soluble AH conformations.

We hypothesized that the PEEL-1 AH may perform a similar role as it does in actinoporins, where AHs construct the lining of a toxic channel. To test this, we determined whether this helix was required for toxicity in HEK293T cells and *C. elegans*. We found that deleting the last two helices of PEEL-1 did not severely impair toxicity ("−28 aa" and "−39aa" in Fig 4C–4D). However, deleting the AH resulted in complete loss of toxicity in both mammalian cells and worms ("−65 aa"

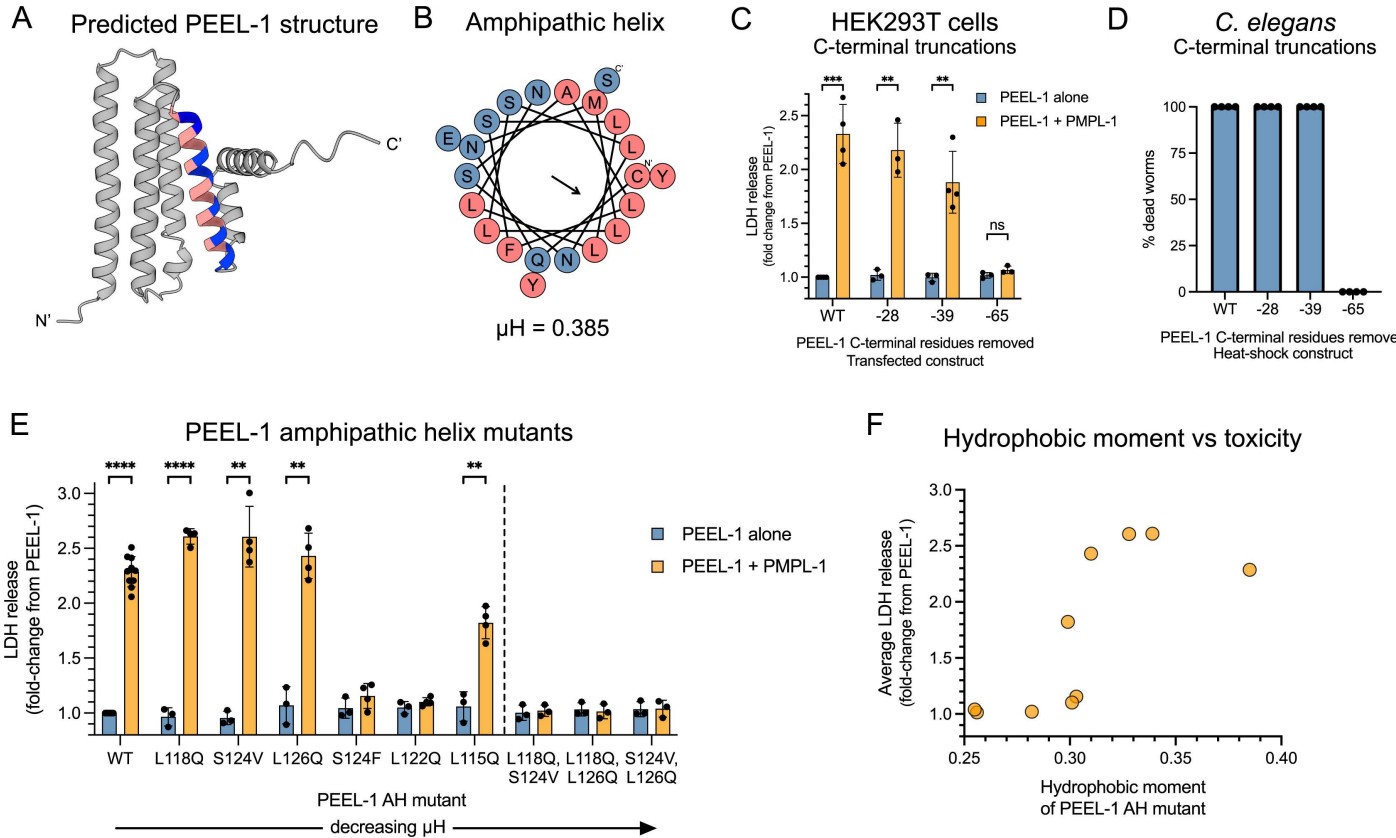

**Fig 4. PEEL-1's amphipathic helix is critical for toxicity. (A)** The AlphaFold2 predicted structure of PEEL-1 and **(B)** a helical wheel representation of the putative PEEL-1 amphipathic helix. Amphipathic helix residues are colored (pink = hydrophobic, blue = hydrophilic). **(C)** Cytotoxicity of a series of PEEL-1 C-terminal truncations expressed in HEK293T cells. The number of amino acids removed are indicated (ex. "−28" means the last 28 residues were removed). Each truncation removes an additional alpha helix. The "−65" truncation removes the amphipathic helix. **(D)** Percent dead worms after heat-shock PEEL-1 expression of the indicated truncation mutant. Fifty worms were assayed for each data point, and two independent transgenic lines were tested for each construct. **(E)** Cytotoxicity of PEEL-1 amphipathic helix missense mutants in HEK293T cells. Mutants are ordered by descending hydrophobic moment (μH). Six single mutants (left of dotted line) and three double mutants are shown (right of dotted line). All bar graphs show mean with SD. Statistics were performed using multiple unpaired *t*-tests with Holm-Šídák test, comparing each PEEL-1 alone to PEEL-1 and PMPL-1 (**$p < 0.01$; ***$p < 0.001$; ****$p < 0.0001$). **(F)** The average toxicity of missense mutants from (E) plotted against their hydrophobic moment. Underlying data are available in S1 Data.

in Fig 4C–4D). By deleting one amino acid at a time, we determined that toxicity in HEK293T cells was eliminated after removing at least 40 amino acids (S10 Fig). This PEEL-1 (−40aa) mutant contains only three residues following the AH, possibly destabilizing the AH.

We also tested whether the amphipathic property of the PEEL-1 AH was critical for toxicity by creating a series of missense mutants that modified the AH's hydrophobic moment. Six single-missense mutants were tested. Three of these mutants attenuated PEEL-1 toxicity (S124F, L122Q, L115Q) while three mutants were still fully toxic (L118Q, S124V, L126Q) (Fig 4E). However, all pairwise combinations of these three fully toxic mutants resulted in complete loss of toxicity (Fig 4E). Across the nine missense mutants we tested, toxicity correlated well with the amphipathicity of the AH (Fig 4F). These data suggest that the amphipathic property of the PEEL-1 amphipathic helix is critical for toxicity, supporting its potential role as the lining of a channel. However, we cannot rule-out alternative explanations for the importance of the AH, including roles in protein–protein interaction or protein stability.

## PEEL-1 and PMPL-1 create a non-specific monovalent cation channel

We hypothesized that PEEL-1 toxicity was caused by disruption of ionic gradients across the cell membrane, leading to osmotic imbalance and subsequent cell swelling. This could be through two mechanisms: an ion leak channel, where specific ions flow down their electrochemical gradient, or a non-selective pore, causing flux of all ions and smaller osmolytes across the plasma membrane. To test these possibilities, we performed whole-cell patch-clamp electrophysiology on HEK293 cells in physiological ion conditions (high internal $K^+$, high external $Na^+$). Currents from cells transfected with *peel-1* or *pmpl-1* alone appeared similar to currents from untransfected cells (Figs 5A and S11A–11B). Obtaining recordings of cells co-transfected with *peel-1* and *pmpl-1* was difficult, due to variable timing of toxicity, and was complicated by loss of cells due to swelling. Therefore, we generated tetracycline-inducible cell lines that more tightly regulate PEEL-1 and PMPL-1 expression. We generated a HEK293 cell line with tetracycline-inducible PMPL-1::mCherry and then introduced stable expression of either eGFP ("Control" cell line) or PEEL-1::eGFP ("Experimental" cell line). Cells with and without tetracycline are hereafter referred to as induced and uninduced, respectively. Nearly all Experimental cells swell 24 hrs after induction (S12A Fig), with the first signs of swelling occurring 6 hrs after induction (S12B Fig).

We performed whole-cell patch clamp recordings without leak subtraction of uninduced cells, and at 6−7 hrs and 28 hrs after induction (Fig 5B–5C). Similar to untransfected naïve HEK293 cells, uninduced Control cells exhibited small instantaneous negative (inward) currents at negative potentials, and these currents did not increase after induction (Figs 5B and S11C). In contrast, uninduced Experimental cells exhibited a larger instantaneous inward current compared to uninduced Control cells and naïve HEK293 cells, and the magnitude of these currents further increased after 28 hrs of induction (Figs 5C and S11C). When clamped at the most negative potential (−100 mV), inward currents from uninduced and 28-hr induced Experimental cells were 7-fold and 30-fold greater than naïve HEK293 cells, respectively. These data suggest that co-expression of PEEL-1 and PMPL-1 creates a constitutively active ion channel.

The inward currents observed in uninduced Experimental cells were instantaneous, exhibiting no voltage-dependent gating kinetics (Figs 5C and S11C), consistent with a leak channel constitutively open at physiological resting membrane potentials. This ion leakage in uninduced Experimental cells was surprising since these cells appear healthy and do not have obvious growth defects. We detected some PMPL-1 expression in uninduced Control and Experimental cells (Figs 5D and S12A), consistent with previous reports of leaky background expression from tetracycline-inducible promoters [30–32]. Therefore, we reasoned that uninduced Experimental cells may have sufficient background PMPL-1 expression to display an electrophysiological phenotype but not enough to cause cell swelling or cell death.

To characterize the ionic selectivity of channels formed by PEEL-1 and PMPL-1 co-expression, we performed ion substitution experiments. We took advantage of the inward currents generated by uninduced Experimental cells since these cells are morphologically normal, reducing the possibility of contamination by endogenous HEK293 channels secondarily activated by cell swelling or cell death such as the LRRC8-associated volume-regulated anion channels (VRACs) [33,34]. We substituted the physiological mixed cation bath solution (high $Na^+$/low $K^+$) with equivalent bath solutions containing single cations ($Na^+$, $K^+$, $Cs^+$, $NMDG^+$ or a combination of 20 mM $Ca^{2+}$/120 mM $NMDG^+$) and monitored instantaneous inward currents at negative potentials. We found that bath solutions with $Na^+$, $K^+$, or $Cs^+$ still yielded large inward currents, while bath solutions with $NMDG^+$ and $Ca^{2+}$ abolished all inward currents (Figs 5E and S13). These results suggest that PEEL-1 and PMPL-1 co-expression creates a channel that is permeable to monovalent cations ($Na^+$, $K^+$, and $Cs^+$) but impermeant to the divalent $Ca^{2+}$ cation and the bulkier cation $NMDG^+$. We next tested for $Cl^-$ permeability by recording with high internal $Cl^-$ (140 mM KCl) and low external $Cl^-$ (140 mM $NMDG^+$). Any $Cl^-$ permeability at negative potentials would cause $Cl^-$ movement out of the cell and be recorded as a negative (inward) current. We observed no inward currents (Figs 5E and S13), indicating that the channel is impermeable to $Cl^-$. Altogether, these results suggest that PEEL-1 and PMPL-1 create an ion channel rather than a non-selective pore. The PEEL-1 and PMPL-1 channel conducts monovalent cations and is impermeable to anions and divalent cations. An increase in PMPL-1 expression in induced Experimental cells may

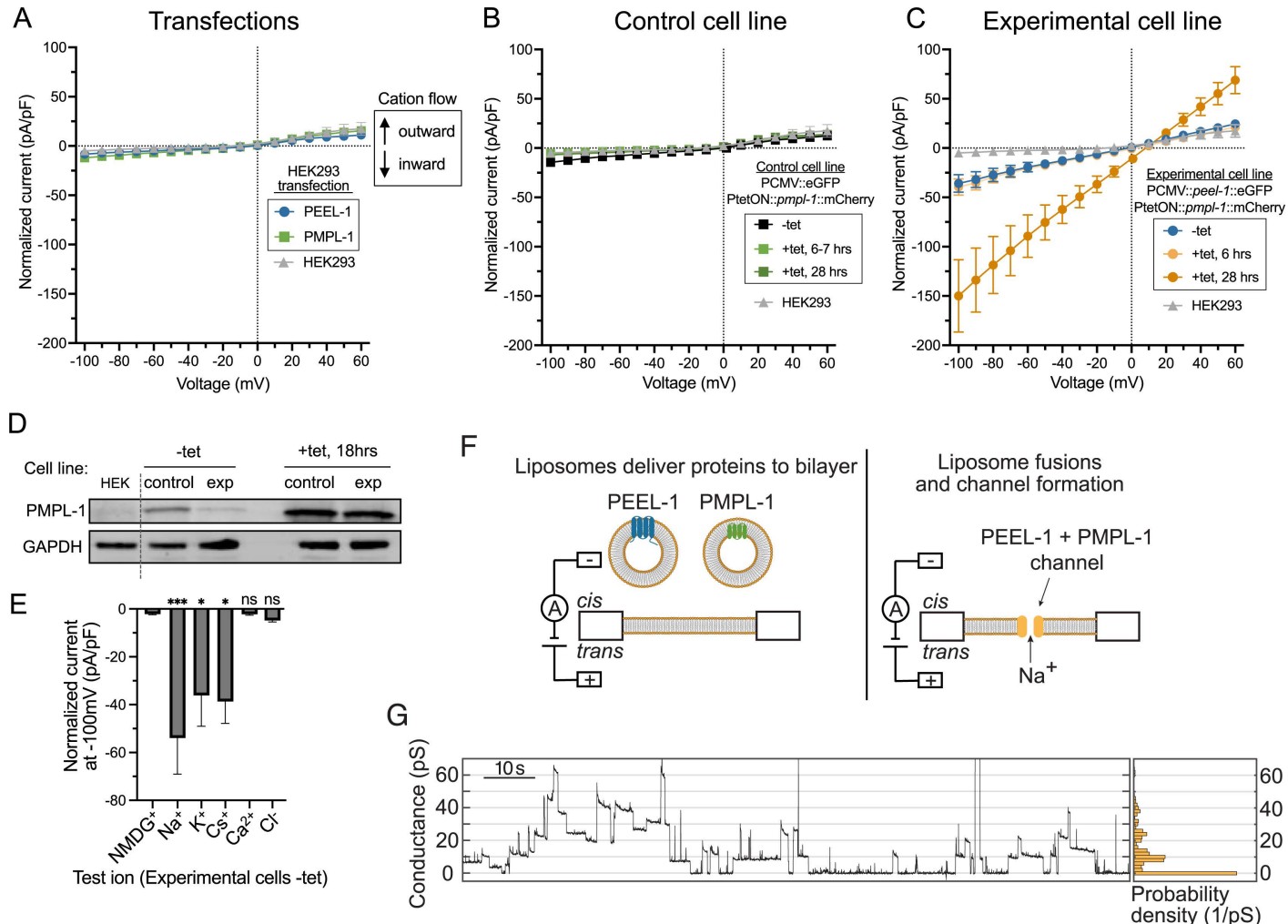

**Fig 5. PEEL-1 and PMPL-1 create a monovalent cation channel.** Current–voltage plots of whole-cell patch-clamp electrophysiology on **(A)** trans-fected cells, **(B)** Control cell line, and **(C)** Experimental cell line. High intracellular potassium (140 mM $K^+$/8.6 mM $Na^+$) and high extracellular sodium (145 mM $Na^+$/4 mM $K^+$) solutions are used. Currents elicited by a family of 0.5 s voltage steps from a −30 mV holding potential, from −100 mV to 60 mV, in 10 mV increments. Currents normalized to cell capacitance (pF). Negative and positive currents indicate cation flow into or out of the cell, respectively. **(A)** Plots from HEK293 cells acutely transfected with *peel-1*::eGFP or *pmpl-1*::mCherry. **(B)** Plots from Control cell line expressing constitutive eGFP and tetracycline-inducible *pmpl-1*::mCherry. **(C)** Plots from Experimental cell line expressing constitutive *peel-1*::eGFP and tetracycline-inducible *pmpl-1*::mCherry. Control and Experimental cell lines are shown without tetracycline or with tetracycline at the indicated time after addition of tetracy-cline. All plots shown as mean with SEM. **(D)** Western blot for PMPL-1::mCherry in Control and Experimental cell lines without tetracycline (−tet, left) and 18 hrs after addition of tetracycline (+tet, right). Leaky expression of PMPL-1 is seen in both cell lines in the absence of tetracycline. Less background PMPL-1 expression is seen in Experimental cells than in Control cells, likely because of selection against higher background PMPL-1 expression when in combination with PEEL-1 but not eGFP. GAPDH loading control shown. Original blots available in S1 Raw images. **(E)** Permeability of indicated ions was assayed in Experimental cells without tetracycline. Test ionic solutions substituted previous bath solution (145 mM $Na^+$/4 mM $K^+$) with 140 mM pure cations (external) or 20 mM anions (internal), except $Ca^{2+}$ (20 mM external, 1 mM internal $Ca^{2+}$ with 2.5 mM EGTA). All plots show mean with SEM. Statistical tests compare all results to NMDG (treated as control) in one-way ANOVA with Dunnett's multiple comparisons test (*$p < 0.05$; ***$p < 0.001$). **(F)** Schematic of planar lipid bilayer experiment. Liposomes containing purified PEEL-1 or PMPL-1 are added to the *cis* well to deliver proteins to the lipid bilayer. **(G)** Conductance traces of one experiment (left) and a histogram of the trace (right, 2 pS bin width, normalized based on probability density) at an applied voltage of +180 mV. SDS–PAGE gels of purified proteins are shown in S15A Fig and more example conductance traces and controls are shown in S16 Fig. Underlying data are available in S1 Data.

result in excessive influx of Na⁺, sufficient to overwhelm compensatory volume regulatory mechanisms, leading to osmotic dysregulation, cell swelling, and cell death.

## Predicted PEEL-1 pentamer structure has features of an ion channel

Ion channels are typically made up of oligomeric complexes, so we used AlphaFold2 to determine possible oligomeric structures of PEEL-1. Structural predictions were of low-confidence due to PEEL-1's lack of homology to known proteins [5]. Nevertheless, the predicted PEEL-1 pentamer has a striking resemblance to known cation channels (S14A–14D Fig and S3 Data): (a) the outside surface of the complex is hydrophobic, consistent with its location in lipid bilayers, (b) the complex has a central, uninterrupted hydrophilic pore, providing a potential path for ions through the complex, and (c) the opening of the pore region is surrounded by a ring of negatively charged residues. These features are strikingly similar to the structure of the ZAR1 cation channel, which is a toxic pentameric channel with a ring of acidic residues at the extracellular mouth of the pore that serve as a cation selectivity filter [35]. In the predicted PEEL-1 pentamer, a similar ring of negative charges is formed by five D109 residues. Mutating this residue to an alanine (D109A) resulted in complete loss of PEEL-1 toxicity (S14E Fig). This residue is two amino acids before the PEEL-1 AH. Five copies of the PEEL-1 AH meet in the middle of the complex, forming the pore-like region of the predicted structure (S14A–14B Fig), consistent with our model of PEEL-1 AHs constructing the lining of a channel.

## *In vitro* reconstitution of the PEEL-1/PMPL-1 ion channel

We used an independent approach to test if PEEL-1 and PMPL-1 create an ion channel by assaying whether these purified proteins allow ions to flow through artificial planar lipid bilayers (Fig 5F). PEEL-1 and PMPL-1 were individually expressed in *E. coli* with maltose-binding protein (MBP) tags, affinity purified in detergent, and incorporated into liposomes for delivery to planar lipid bilayers (S15A Fig and "Materials and methods"). For PEEL-1, the full-length protein was purified and incorporated into liposomes, along with co-purified truncation products (S15A Fig). After adding PEEL-1 or PMPL-1 liposomes to the planar lipid bilayer, we observed no currents (S16B–16C Fig). After adding both PEEL-1 and PMPL-1 liposomes to the same bilayer, we observed discrete, step-like conductance events similar to those created by typical ion channels (Figs 5G and S16D–16E) [35–38]. There was variation in single channel conductance and open duration (Figs 5G and S16D–16E). Such variability could reflect heterogenous stoichiometries, or it could be due to technical limitations such as misfolded proteins, truncated PEEL-1 proteins in the sample, mixed orientation of proteins in the bilayer, or imprecise amounts of each protein in the bilayer. Nevertheless, the stepwise conductance *in vitro* and cation-selective property in cells suggest that PEEL-1 and PMPL-1 together construct a monovalent cation channel which causes osmotic dysregulation and cell death.

## Discussion

Our study provides unprecedented understanding of an animal toxin-antidote system. We show that the PEEL-1 toxin co-opts a conserved membrane protein, PMPL-1, to create a cation channel. Opening of this toxic channel likely causes a persistent influx of sodium into the cell which may increase the osmolarity inside the cell. This would in turn cause water movement into the cell and cell swelling. It is unclear how cell swelling leads to cell death. We speculate that there may be a failure in the homeostatic mechanisms that normally counteract cell swelling, ultimately leading to cell lysis. One possible mechanism is that the persistent Na⁺ influx may overwork the ATP-dependent Na⁺/K⁺ pump that acts in transporting Na⁺ back out of the cell. This could lead to ATP depletion and a subsequent failure in the homeostatic mechanisms that regulate cell volume such as activation of the volume-regulated anion channel (VRAC), which requires ATP for its activity. Disruption of other essential ATP-dependent cell processes may also contribute to cell death.

Our data are consistent with a toxic cation channel being constructed by PEEL-1 oligomers, and we hypothesize that the PEEL-1 amphipathic helix creates the hydrophilic lining of the channel. Although we show that PMPL-1 is required

for toxicity, the specific biochemical role of PMPL-1 is unknown. PMPL-1 may interact directly with PEEL-1, either as a structural component of the channel or by inducing conformational changes in PEEL-1 that gate the channel open. Alternatively, PMPL-1 may indirectly promote PEEL-1 activity. For instance, PMPL-1 may modify biophysical properties of lipid bilayers, such as membrane fluidity or curvature, which could subsequently activate the PEEL-1 ion channel. We favor a direct protein-protein interaction hypothesis since PMPL-2 cannot replace PMPL-1 function in toxicity, despite its high amino acid identity.

We show that that purified PEEL-1 and PMPL-1 proteins together are sufficient for ion flux through planar lipid bilayers, and that neither protein on its own produces these currents. These planar lipid bilayer experiments are preliminary and further work will be required to fully characterize the ion channel activity. In future work, this *in vitro* system could be used to further characterize the biophysical properties of the channel such as its ion selectivity and single-channel conductance. Structural hypotheses can also be tested *in vitro* by assaying the properties of purified mutant proteins. However, purification of PEEL-1 is a challenge in our current system since this protein is toxic when overexpressed in *E. coli* (S15B Fig), resulting in low yields and truncated products. Purifying PEEL-1 from mammalian cells is a possible solution to this problem, since we have shown that PEEL-1 alone is not toxic to HEK293T cells.

We found that ZEEL-1 is sufficient for antidote activity in HEK293T cells, indicating that this heterologous system can also be used to test mechanistic hypotheses of antidote activity. ZEEL-1 is a predicted transmembrane protein with a soluble domain homologous to a substrate-recognition subunit of an E3 ubiquitin ligase [4]. We hypothesize that PEEL-1 and ZEEL-1 interact in the membrane, resulting in ubiquitylation of PEEL-1 and subsequent degradation. The core E3 ligase machinery is conserved, and other *C. elegans* E3 subunits are functional in mammalian cells [39]. Thus, ZEEL-1 may act in HEK293T cells with endogenous mammalian ubiquitin ligase machinery to prevent PEEL-1 toxicity.

Our work highlights a possible new theme in animal toxin-antidote systems, in which the timing of toxicity may be controlled by co-opted, endogenous proteins. Sperm deliver PEEL-1 protein to offspring, but sperm lack the antidote ZEEL-1. Thus, PEEL-1 toxicity must somehow be avoided in sperm cells. Toxicity must also be avoided in early embryos that lack ZEEL-1, as ZEEL-1 is only expressed in embryos after the initiation of zygotic transcription, beginning at the eight-cell stage [5]. If PEEL-1 was toxic to embryos before zygotic gene expression, *zeel-1(+)* progeny would die, breaking the selfish activity of *peel-1 zeel-1*. One solution to controlling the timing of toxicity could be the requirement for PMPL-1. Cells lacking PMPL-1 protein would be immune to PEEL-1 toxicity, so we expect PMPL-1 to be absent in sperm and early embryos. All animal toxin-antidote systems must solve the paradoxical problem of how to deliver a toxic molecule through the germline to the embryo while only allowing toxicity in later embryonic stages after the maternal-to-zygotic transcriptional switch. Control of toxin activity by requiring a second, endogenous protein that is absent in gametes and early embryos may be a solution shared by many animal toxin-antidote systems.

## Materials and methods

### Worm strains and maintenance

Worm strains were maintained using standard procedures [40]. Strains used in this study are provided in S1 Table.

### Isolation of suppressors of heat-shock induced PEEL-1

Worm strains XZ1047 and XZ1372 were mutagenized using ENU or EMS [40,41]. These strains contain two copies of *hsp-16.41p*::*peel-1* in order to avoid isolating suppressors that have mutations in the transgene. F2 populations of worms were heat-shocked at 34 °C for 2 hrs and allowed to recover at room temperature overnight. Surviving worms were isolated. Approximately 200,000 haploid genomes were screened. *yak52* was isolated from EMS mutagenesis of XZ1047, while *yak103* was isolated from ENU mutagenesis of XZ1372.

We isolated a total of six independent suppressors of heat-shock PEEL-1 toxicity. Four of the mutations were partial suppressors while two were full suppressors. The causative genes were identified using genetic mapping and

whole-genome sequencing. The four partial suppressors have mutations in mRNA export factors *nxf-1* and *nxt-1*, but act by affecting expression of *hsp-16.41p::peel-1* and do not suppress toxicity of endogenous, sperm-delivered PEEL-1 [42].

The two full suppressors *yak52* and *yak103* are recessive suppressors of PEEL-1 toxicity and have no other obvious phenotype. They were determined to be allelic by complementation testing and were mapped to the X chromosome by crossing to strains carrying visible or fluorescent markers on each chromosome (strains EG1000, EG1020, EG8040, EG8041). The closest linkage seen was to the visible marker *lon-2*.

### Identification of *pmpl-1*

Strains XZ1177 and XZ1307 carrying 4X and 5X outcrossed *pmpl-1(yak52)* and *pmpl-1(yak103)* were subjected to whole-genome sequencing. DNA was purified according to the Hobert laboratory protocol (http://hobertlab.org/whole-genome-sequencing/). Illumina (XZ1177) or Ion Torrent sequencing (XZ1307) was performed at the University of Utah DNA Sequencing Core Facility. The dataset for XZ1177 contained approximately 17,000,000 reads of a mean read length of 36 bases, resulting in approximately 6X average coverage of the *C. elegans* genome. The dataset for XZ1307 contained approximately 12,800,000 reads of a mean read length of 147 bp, resulting in approximately 19X average coverage. The sequencing data were processed on the Galaxy server at usegalaxy.org [43]. SNPs and indels were identified and annotated using the Unified Genotyper and SnpEff tools [44,45]. Although we found genes on the X chromosome in each strain containing nonsynonymous mutations, none of these genes were affected independently in both strains. Thus, to identify potential deletions, we used BEDtools Genome Coverage tool to calculate sequencing coverage across the genome in intervals of consecutive bases with the same coverage [46]. Intervals with zero sequencing coverage were then annotated using SnpEff, filtered to intervals on the X chromosome which overlapped protein-coding regions, and sorted by length. In XZ1307, the largest of these intervals on the X chromosome was found near where *yak103* and *yak52* had been mapped and was subsequently confirmed to correspond to *yak103*, a 323-bp deletion spanning the F47B7.1 gene, from its 5′ UTR to 3′ UTR and deleting all the coding sequence. In XZ1177, we found two small regions of F47B7.1 lacking coverage in our whole genome sequencing. Via Sanger sequencing, we identified *yak52* as a nonsynonymous mutation in one of these regions, with a G to A mutation that results in an A47T substitution.

### Heat-shock PEEL-1 toxicity assay

Heat-shock PEEL-1 assays were performed with 50 gravid adults of each strain for each biological replicate. Adults were picked to NGM plates seeded with OP50 and heat-shocked (2 hrs at 34 °C). After recovering at room temperature for 2 hrs, moving worms were counted. Non-moving worms were tested for response to touch stimulus and scored as dead if they did not move.

### Sperm-delivered PEEL-1 toxicity assay

Sperm-delivered PEEL-1 toxicity in wild-type or *pmpl-1* mutant backgrounds was assayed by counting unhatched embryos. To make *pmpl-1; peel-1(+) zeel-1(+)/peel-1(−) zeel-1(−)* worms (Fig 1C), AFS216 *peel-1(−) zeel-1(−)* males were crossed to *peel-1(+) zeel-1(+)* worms with *pmpl-1* mutations *yak103* (XZ2194), *yak52* (XZ2283), or wild-type *pmpl-1* (N2, control). F1 males were backcrossed to the parent strain (XZ2194, XZ2283, or N2) to obtain the desired genotype. To determine if *pmpl-1* acts by a paternal effect (S1 Fig), F1 males were backcrossed instead to AFS216 *peel-1(−) zeel-1(−)*.

Embryonic lethality was assayed on individual, self-fertilizing hermaphrodites. Single worms were allowed to lay embryos for 16−24 hrs before being removed from the plate. Plates were left for one more day to allow embryos to develop and hatch, and the progeny were scored. Unhatched embryos and larval worms were counted. Unhatched embryos were scored as dead. Progeny were genotyped in bulk for *zeel-1(−)* by PCR using oGP65 (5′ attctggagttcgtgaggtgc 3′) and oGP66 (5′ ccctcctttcccacccaac 3′). Only plates showing *zeel-1(−)* alleles were considered to have parents with the desired genotype, heterozygous *peel-1(+) zeel-1(+)/peel-1(−) zeel-1(−).*

## Plasmid construction

All plasmids used in this study are provided in S2 Table. Plasmids used for mammalian cell transfection were generated using Gibson assembly [47] using N1 vector backbone (CMV promoter). Constructs for tetracycline-inducible expression were cloned into pFTSH vector (gift from Nancy Maizels). Plasmids used for *C. elegans* transgenics were constructed using either Gibson assembly or Gateway cloning (Invitrogen). Typical Gateway cloning was performed using a 3-fragment approach (promoter, coding region, and fluorophore-UTR or UTR) into a destination vector, pCFJ150. Restriction digest was used to confirm the plasmid was correct. Sanger sequencing was used to confirm that constructs did not contain mutations (S2 Table). Transgenic worms were made using microinjection. For each transgenic strain, we isolated at least two independent lines and confirmed similar results.

## Microscopy

All microscopy was performed on a NikonTi2-E Crest X-light V2 spinning-disk confocal microscope. For live-cell imaging, mammalian cells were maintained in a humidified, heated chamber (37 °C) in 5% carbon dioxide. For long-term imaging, the objective heater was also heated to 37 °C. For imaging *C. elegans*, worms or embryos were picked onto a 2% agar pad, immobilized in a humidified chamber for 10 min in 33 mM sodium azide, and then immediately imaged.

## Mammalian cell maintenance and generation of clonal cell lines

All mammalian cells were grown at 37 °C and 5% carbon dioxide in DMEM with GlutaMAX (Thermo Fisher Scientific), 10% fetal bovine serum (RMBIO or Gibco), and 100 U/mL of penicillin-streptomycin (Gibco). All transfection-based cytotoxicity experiments were done in HEK293T cells (gift from Mary Claire-King). For electrophysiology experiments, HEK293 cells (CRL-1573; ATCC) were used. Tetracycline-inducible cells (tetON cells) were made in HEK293 Flp-In TReX cells (Invitrogen) (gift from Nancy Maizels). tetON cells were grown in media made with Tet system approved fetal bovine serum (Gibco).

Stable tetON lines were generated by co-transfection of FlpO (pOG44; Life Technologies) and PtetON::*pmpl-1*::mCherry (pLC79, in pFTSH backbone, a gift from Nancy Maizels). Two days after transfection, cells were plated at low density in 10-cm plates with hygromycin B (150 µg/mL). Two clonal populations were picked into media with hygromycin B (150 µg/mL), grown to 80% confluency, and frozen in 10% DMSO at −80 °C. One clonal population was used to generate the Experimental and Control cell lines.

Experimental tetON and Control tetON cell lines were generated in parallel. One tetON::*pmpl-1*::mCherry clonal cell line was transfected with the respective plasmid, *peel*-1::eGFP (pGP9) or eGFP. Two days after transfection, cells were plated at low density in selective media containing both hygromycin B (150 µg/mL) and G418 (400 µg/mL). After about two weeks, multiple single colonies were picked and continued to be grown under selection. For the Experimental cell line, more than half the picked clones which grew in selective media did not have visible green fluorescence, presumably due to leaky *pmpl-1*::mCherry imposing selection against *peel-1*::eGFP-expressing cells. Therefore, only clones which had green fluorescence were maintained. This issue was not encountered during the generation of the Control cell line. All Experimental and Control cell lines were screened following tetracycline treatment. Forty-eight hours after tetracycline treatment, all cell lines had red fluorescence, all Experimental cell lines had very few adhered cells (*n* = approximately 6), and all Control cell lines had no obvious cellular phenotypes (*n* = approximately 10).

## Cytotoxicity assay

Cytotoxicity in mammalian cells was measured using a colorimetric assay for LDH release (Promega CytoTox 96). 12-well plates were seeded with approximately $7 \times 10^4$ HEK293T cells about 24 hrs prior to transfection. Each well was transfected with a mixture of 1.5 µg DNA and 3 µg polyethylenimine (PEI) in 200 µL OptiMEM (Gibco). Three constructs were

transfected in all experiments. For each of PEEL-1, PMPL-1, and ZEEL-1, we either transfected a fluorophore-tagged protein construct or the fluorophore-only vector as a control. Supernatant was collected 43−45 hrs after transfection and used in 96-well plates for LDH assays, following manufacturer protocol. Two technical replicates were measured for each well and absorption at 490 nm was averaged between replicates. All experiments included wells for a non-killing control (fluorophore-encoding vectors), *peel-1*::eGFP alone, and *peel-1*::eGFP with *pmpl-1*::mCherry. The fold-change over *peel-1*::eGFP was calculated from transfections performed on the same day.

## SDS–PAGE and western blots

For lysis of mammalian cells, cells were collected from 12-well plates and washed with PBS. Cell pellets were lysed in radioimmunoprecipitation assay (RIPA) lysis buffer (25 mM Tris pH 7.4, 150 mM NaCl, 0.1% SDS, 1% NP-40, 1% sodium deoxycholate) with DNase and Halt Protease Inhibitor Cocktail (Thermo Fisher Scientific). After lysis (15 min on ice), the sample was centrifuged (21,000 × $g$, 10 min, 4 °C), and the supernatant was used for SDS–PAGE.

Following separation by SDS–PAGE, proteins were transferred onto a nitrocellulose membrane. After blocking in Intercept Blocking Buffer (LI-COR, 1 hr room temperature) membranes were incubated with primary antibody and nutated overnight at 4 °C. Mouse monoclonal anti-mCherry (a gift from Jihong Bai and the Fred Hutch Cancer Center antibody development shared resource center; 1:1000), mouse monoclonal anti-MBP (New England Biolabs, E8032; 1:10,000), rabbit polyclonal anti-GAPDH (Sigma Aldrich, G9545; 1:5,000). Appropriate secondary antibodies were used, either goat anti-mouse or donkey anti-rabbit conjugated to Alexa 680 or Alexa 790 (Invitrogen). Membranes were imaged on an Odyssey CLx (LI-COR Biosciences).

## Electrophysiology

For electrophysiology on acutely transfected cells, plasmid constructs were transfected into HEK293 cells using Viafect reagent (Promega), following the manufacturer's protocol. Cells were first prepared for transfection by plating onto 12-well tissue culture plates (Nunc 12-565-321; Thermo Fisher Scientific) at a density of approximately $0.5−2 × 10^5$ cells per well and grown to approximately 80%–90% confluence with standard media, allowing for one confluent well per transfection condition. On the day of transfection, media were replaced with 0.5 mL fresh DMEM with 10% fetal bovine serum (Gibco), without penicillin-streptomycin. Lipophilic/DNA transfection complexes were generated for each well, combining a total of approximately 1.0 µg of plasmid DNAs with serum-free OptiMEM (Gibco) to a final volume of 100 µL, then adding 3.0 µL Viafect with gentle trituration, allowing the mixture to assemble at 24 °C for 30 min, and then added to each well, dropwise. Transfected cells were incubated overnight at 37 °C and visually monitored for transfection efficiency in situ using an inverted plate microscope equipped with fluorescence (Invitrogen EVOS M7000; Thermo Fisher Scientific). Transfection efficiencies were typically >70%–80%. Specific amounts of plasmid DNAs for acute transfections per well: (a) pCMV::*peel-1*::eGFP (0.8 µg), (b) pCMV::*pmpl-1*::mCherry (0.1 µg) and pcDNA3 (0.8 µg), (c) pCMV::*peel-1*::eGFP (0.8 µg) and pCMV::*pmpl-1*::mCherry (0.1 µg). Untransfected HEK293 cells served as controls.

Following overnight incubation, cells in transfected wells were dissociated with TrypLE (Gibco); and replated at low density onto 12 mm poly-*D*-lysine-coated glass coverslips (NeuVitro) in 24-well tissue culture plates (FisherBrand FB012929; Thermo Fisher Scientific), for patch-clamp electrophysiology. Typically, approximately 10,000–15,000 cells were replated per well at sufficiently low density to isolate individual cells. This was necessary to prevent the formation of electrical junctions between contacting cells, which precludes adequate space-clamp recording conditions. Recordings were performed from 0.5 to 3 days after replating at low densities. Control and Experimental stable HEK293 cell lines were similarly replated at low density on coverslips for patch-clamp recordings.

For patch-clamp recordings, coverslips containing adherent cells were transferred to a Zeiss AxoExaminer.A1 microscope, equipped with a 40X water immersion objective and epifluorescence capability. Pipettes were positioned with a

Sutter MPC-325 micromanipulator (Novato). Whole-cell voltage-clamp recordings were acquired with an AxoClamp200B amplifier (Molecular Devices), using pClamp10.4. Composition of recording solutions are listed below:

**Bath solutions.**

1. *HEK293 bath (4 $K^+$, 145 $Na^+$) (in mM)*: 4.0 KCl, 145 NaCl, 2.0 $CaCl_2$, 2.0 $MgSO_4$, 10 HEPES, 10 glucose, pH to 7.4 with NaOH.

2. *140 $Na^+$ (in mM):* 140 NaCl, 2.0 $CaCl_2$, 2.0 $MgCl_2$, 10 HEPES, pH to 7.4 with NaOH.

3. *140 $K^+$ (in mM):* 140 KCl, 2.0 $CaCl_2$, 2.0 $MgCl_2$, 10 HEPES, pH to 7.4 with KOH.

4. *140 $Cs^+$ (in mM):* 140 CsCl, 2.0 $CaCl_2$, 2.0 $MgCl_2$, 10 HEPES, pH to 7.4 with CsOH.

5. *140 $NMDG^+$ (in mM):* 140 N-methyl-*D*-glucamine ($NMDG^+$), 2.0 $CaCl_2$, 2.0 $MgCl_2$, 10 HEPES, pH to 7.4 with HCl.

6. *20 $Ca^{2+}$ (in mM):* 20 $CaCl_2$, 120 $NMDG^+$, 2.0 $MgCl_2$, 10 HEPES, pH to 7.4 with HCl.

**Internal pipette solutions.**

1. *140 $K^+$, low $Cl^-$ (in mM)*: 140 K-*D*-gluconate, 1.0 $CaCl_2$, 2.0 $MgCl_2$, 10 HEPES, 2.4 EGTA, 4.0 $Na_2ATP$, 0.3 $Na_2GTP$, pH to 7.4 with KOH.

2. *140 $K^+$, high $Cl^-$ (in mM)*: 140 KCl, 1.0 $CaCl_2$, 2.0 $MgCl_2$, 10 HEPES, 2.4 EGTA, 4.0 $Na_2ATP$, 0.3 $Na_2GTP$, pH to 7.4 with KOH.

For all recordings of cationic currents, different bath solutions were used in combination with *140 $K^+$, low $Cl^-$* internal pipette solution. For recordings of $Cl^-$ currents, *140 $NMDG^+$* bath solution was used in combination with *140 $K^+$, high $Cl^-$* internal pipette solution; under these conditions outward $Cl^-$ conductance at negative potentials would be seen as inward currents by standard electrophysiological recording convention. Patch pipettes were pulled from borosilicate glass (1B120F-4; World Precision Instruments) on a P-97 Sutter Instruments puller (Novato), with resistances of 3.0–5.0 MΩ. Currents were allowed 3–5 mins to stabilize after achieving whole-cell recording configuration, filtered at 5 kHz, and acquired at 10 kHz. Series resistance compensation was >80% for all recordings.

Mean currents normalized to cell capacitance were analyzed and plotted using pClamp10.4 (Molecular Devices), Microsoft Excel, and OriginPro8.5 (Northampton). Leak subtraction correction was not applied to any of the data. Current–voltage (I/V) data were plotted as means with standard errors (SEMs) and statistical calculations were performed in Prism (GraphPad).

## Protein purification

MBP::PEEL-1::His$_8$ and MBP::PMPL-1 were purified from *E. coli* BL21(DE3) and C43(DE3) cells, respectively. An overnight culture (37 °C, 220 rpm, 100 µg/mL ampicillin, 33 µg/mL chloramphenicol) was used to inoculate 1 L of LB media (100 µg/mL ampicillin, 220 rpm, baffled flask), grown at 37 °C to $OD_{600}$ of 0.4–0.7, and induced with IPTG (0.5 mM). PMPL-1 cells were chilled on ice prior to induction, and then induced in an 18 °C shaker overnight. PEEL-1 cells were induced at 37 °C for 1 hr since longer induction at lower temperatures resulted in decreased yields. Cells were pelleted and either stored at −80 °C or used immediately for purification.

For purification, cell pellets were resuspended in Buffer A (HEPES pH 7.4, 150 mM NaCl, 5 mM 2-mercaptoethanol, 10% glycerol). Cells were lysed using a Dounce homogenizer, followed by 30 min rocking at room temperature after addition of DNase, lysozyme, protease cocktail inhibitor (Pierce), and n-octyl-β-glucopyranoside detergent (β-OG, 2% final concentration) (Anatrace). Lysate was clarified by centrifugation (20 min, 20,000 × *g*, 4 °C) before batch binding with amylose resin for 1–2 hrs at 4 °C. Resin was pelleted (5 min, 700 × *g*) and resuspended in wash buffer to load onto a

clean column. Four washes (5 column volumes) and four elutions (two column volumes) were performed by gravity flow. Wash buffer and elution buffer contained 1% β-OG in Buffer A. Maltose (10 mM) was included in elution buffer. Protein yield was estimated by absorbance at 280 nm. Proteins were stored at 4 °C for less than three days before incorporation into liposomes.

## Proteo-liposomes

Liposomes containing MBP::PEEL-1::His$_8$ or MBP::PMPL-1 were made with 50% DOPC:POPS (1,2-dioleoyl-sn-slycero-3-phosphocholine: 1-palmitoyl-2-oleoyl-sn-glycero-3-phospho-L-serine). Lipid stocks were purchased in chloroform (Avanti Polar Lipids) and β-OG stock was made in methanol. Lipid films were made by mixing lipids with β-OG detergent (lipid:detergent molar ratio of 4:35) in glass vials, dried under a nitrogen stream (10–20 min), and further dried in a Speedvac evaporator (4–6 hrs). Buffer A was added to the dried lipid-detergent mixture and resuspended via two or three rounds of bath sonication (5 min, room-temperature) with nutation (20 min, 4 °C). Protein was added in an approximate protein:lipid molar ratio of 1:400 (PMPL-1) or 1:2,500 (PEEL-1). Detergent was removed by dialysis: 500 μL of sample was dialyzed (20 kDa cutoff) in 250 mL Buffer A with 0.5 g Biobeads SM-2 (Bio-Rad) for 16–18 hrs at 4 °C. Dialyzed sample was floated through a density gradient (Histodenz 35%, 25%, 0%; Sigma-Aldrich) using ultracentrifugation (SW60Ti, 55 krpm, 4 °C, 30 min). Liposomes were collected from the top fraction (200 μL, using a wide-bore pipette tip), aliquoted, flash frozen in liquid nitrogen, and kept at −80 °C. Liposomes were thawed fresh on the day of each experiment. A second liposome prep was made from an independent protein purification with two adjustments: soybean lipids (Avanti Polar Lipids) with cholesterol were used for liposomes instead of DOPC and POPS, and 1 mM EDTA was included in Buffer A.

## Synthetic planar lipid bilayers

Two approximately 50 μL wells (*cis* and *trans*) linked by an approximately 20 μm PTFE aperture, were filled with Buffer B (20 mM HEPES, pH 7.4, 150 mM NaCl, 10% glycerol, 1 mM CaCl$_2$). A mixture of asolectin lipid (Sigma–Aldrich) and hexadecane was painted across the aperture to establish a planar bilayer membrane as previously described [48]. To promote liposome fusions, an osmotic gradient across the bilayer was established by perfusion of Buffer C (20 mM HEPES, pH 7.4, 600 mM NaCl, 3.36% glycerol, 1 mM CaCl2) into the *cis* well. A voltage of −180 mV was applied across the membrane by two Ag/AgCl electrodes and the ion current was measured. Liposomes were added to the *cis* well and mixed via pipetting. Sharp transient spikes in currents were interpreted as successful liposome fusions. After observing channel activity, Buffer D was perfused into the *cis* chamber (20 mM HEPES, pH 7.4, 600 mM NaCl, 10% glycerol) to remove free liposomes and a voltage of +180 mV was applied since channel activity was more stable at +180 mV than −180 mV. Ion current was recorded at 50 kHz and down sampled to 50 Hz for analysis. The recorded ion current is divided by the applied voltage to give units of conductance. All-points histograms of conductance were constructed using a bin width of 2 pS and normalized based on probability density (area under the histogram equals 1).

We observed channels in 8 independent experiments and with two independent protein preps. A subset of experiments was run with controls, where we observed ion channels in 2 out of 3 trials after adding both PEEL-1 liposomes and PMPL-1 liposomes, but we did not see channel activity when adding PEEL-1 alone (*n* = 4) nor PMPL-1 alone (*n* = 3). A typical experiment ran as follows: 2 μL of PEEL-1 liposomes and 2 μL of PMPL-1 liposomes were added to the *cis* chamber and mixed via pipetting. Liposome fusions to the bilayer were confirmed by transient spikes in conductance, beginning approximately 5−20 min after addition and channel activity was seen approximately 5−60 min after fusions began. The amount of time until channels were observed was used as a benchmark for control experiments, where 2 μL of either PEEL-1 liposomes or PMPL-1 liposomes were added to the chamber and allowed to fuse for the same amount of time or longer than experimental runs.

## Supporting information

**S1 Fig.** *pmpl-1(yak103)* **does not act through paternal-effect. (A)** Genetic cross to test for paternal versus zygotic effect of sperm-delivered PEEL-1 suppression by *pmpl-1(yak103)* deletion mutant (denoted *pmpl-1(−)*). Males heterozygous for the selfish element and carrying *pmpl-1(−)* are mated to *p(−) z(−); pmpl-1(+)* hermaphrodites. Suppression via paternal effect would result in approximately 0% embryonic lethality while suppression by a zygotic effect would result in approximately 50% lethality of cross-progeny. **(B)** The percent of dead embryos seen from the cross shown in panel (A). Results are consistent with a model of *pmpl-1(yak103)* suppression of PEEL-1 through zygotic effect. Underlying data are available in S2 Data.
(PDF)

**S2 Fig.** *pmpl-1* **expression pattern in** *C. elegans*. **(A)** Maximum intensity projection of a *C. elegans*, 1.5-fold embryo with GFP driven by the *pmpl-1* promoter and co-injection markers *myo-2p*::mCherry, *rab-3p*::mCherry, and *myo-3p*::mCherry. Toxicity from sperm-delivered PEEL-1 occurs after this embryonic stage. **(B)** Maximum intensity projections of hermaphrodite adult head (top), midbody (middle), and tail (bottom) of the same strain shown in panel (A). **(C)** Heatmap of *pmpl-1* tissue expression scores from RNA-seq dataset of Day 1 adult worms from Kaletsky and colleagues, 2018 [6]. Tissues with the highest *pmpl-1* expression (red) and lowest expression (blue) are shown. Only the 8 most highly expressed tissues (left) and the 8 most lowly expressed tissues (right) are shown. Expression of *pmpl-1* is lowest in the male gonad (arrow). **(D)** *pmpl-1(yak103)* worms with vulval muscle cell expression of *peel-1*::GFP alone (top) or with *pmpl*-1::GFP (bottom). Number of scored worms are indicated. Green channel brightness is increased in the bottom panel to show fluorescence in the vulval muscle, since toxicity in this cell likely caused decreased levels of fluorescent-tagged proteins. The vulval muscle appears swollen in the green channel when co-expressing *peel-1* and *pmpl-1*. This is the same worm as shown in Fig 1E.
(PDF)

**S3 Fig. PMPL-1 is conserved in nematodes.** Alignment of PMPL-1 amino acid sequences (right) between representative *Caenorhabditis* species, *Pristionchus pacificus*, and *Toxocara canis* shown in a species phylogeny (left). The *pmpl-1 (yak52)* allele is mutated at conserved residue A47 (*).
(PDF)

**S4 Fig. 15 PMP3-like proteins in** *C. elegans*. Alignment of all 15 PMP3-like proteins in *C. elegans* identified by BLAST.
(PDF)

**S5 Fig. Structural predictions of PEEL-1 and PMPL-1.** DeepTMHMM predictions of **(A)** PMPL-1 and **(C)** PEEL-1. PMPL-1 is predicted to be a two-pass transmembrane protein. PEEL-1 is predicted to be a four-pass transmembrane protein. Both proteins are predicted to have their N- and C-termini facing the cytosol. Structural predictions from AlphaFold2 of **(B)** PMPL-1 and **(D)** PEEL-1. AlphaFold2 predicts a very high-confidence PMPL-1 structure that suggests it is a monotopic protein (passing through one leaflet of a lipid bilayer). The PEEL-1 structure is of low and very-low confidence, likely because there are no known homologs of this protein.
(PDF)

**S6 Fig. PEEL-1 and PMPL-1 localization. (A–B)** Intestinal cells of an adult *C. elegans* worm with the indicated constructs. **(A)** Wild-type worms expressing *pmpl-1*::tagRFP and GFP::*dgat-2*. DGAT-2 localizes to lipid droplet membranes, and PMPL-1::tagRFP co-localizes to these organelles. Inset shows one lipid droplet. **(B)** *pmpl-1(yak103)* expressing GFP::*dgat-2* and *peel-1*::tagRFP. PEEL-1 signal (arrow) appears on plasma membrane lining the intestinal lumen and does not co-localize with lipid droplets. Autofluorescence from gut granules appears as filled-in circles in both channels (arrowhead). **(C)** HEK293 cells stably expressing *peel-1*::eGFP. PEEL-1::eGFP localizes to the ER and plasma membrane (arrows). Scale bar = 10 μm.
(PDF)

**S7 Fig. Toxicity causes ER swelling, ER fragmentation, and cell lysis.** Live-cell imaging time course of two HEK293T cells transfected with constructs coding for PEEL-1 (untagged), PMPL-1::eGFP, and mCherry::KDEL (ER marker). Imaging began at 20 hrs post-transfection ($t$ = 0 min). The top cell experiences a loss of plasma membrane (PM) integrity at 25 min, followed by ER swelling. The lower cell loses PM integrity at 150 min. Scale bar = 10 μm.
(PDF)

**S8 Fig. Ectopic PEEL-1 toxicity is non-apoptotic.** Percent dead worms after heat-shock induced expression of PEEL-1. Worms deficient in apoptosis (*ced-3*) and cell engulfment (*ced-2* and *ced-5*) were tested. Two independent experiments were done, with $n$ = 50 worms for each data point. Underlying data are available in S2 Data.
(PDF)

**S9 Fig. Amphipathic helix properties of Actinoporin toxins.** Hydrophobic moment (μH) and hydrophobicity (H) of the amphipathic helix of 35 Actinoporin toxin proteins (data acquired from Macrander and Daly, 2016 [29]) (gray circles) and the predicted PEEL-1 amphipathic helix (blue square). Underlying data are available in S2 Data.
(PDF)

**S10 Fig. Toxicity of PEEL-1 C-terminal truncations.** Cytotoxicity of PEEL-1 C-terminal truncations assayed alone (blue bars) or with PMPL-1 (yellow bars) in HEK293T cell transfections. The number of amino acids removed from the C-terminus is indicated (ex. "−28" means 28 amino acids were removed). Data from PEEL-1 WT, −28, −39, and −65 are from Fig 4C. Toxicity is lost upon removal of the −40 residue (Ala135). Statistical tests done using multiple unpaired t-tests with Holm-Šídák test, comparing each PEEL-1 alone to PEEL-1 and PMPL-1 (*$p < 0.05$; **$p < 0.01$; ***$p < 0.001$). Underlying data are available in S2 Data.
(PDF)

**S11 Fig. Raw traces from HEK293 electrophysiology experiments. (A)** Schematic of electrophysiology experiment. High intracellular potassium (140 mM $K^+$/8.6 mM $Na^+$) and high extracellular sodium (145 mM $Na^+$/4 mM $K^+$) solutions are used. Currents elicited by a family of 0.5 s voltage steps from a −30 mV holding potential, from −100 mV to 60 mV, in 10 mV increments. Current traces are normalized to capacitance (pF) and not leak-subtracted. **(B)** Representative traces from untransfected HEK293 cells (top) and HEK293 cells transfected with *peel-1*::eGFP (middle) or *pmpl-1*::mCherry (bottom). Scale bar shown (bottom-right). **(C)** Representative traces of tetracycline-inducible cells lines. Control cells (left) have stable expression of eGFP and inducible expression of *pmpl-1*::mCherry. Experimental cells (right) have stable expression of *peel-1*::eGFP and inducible expression of *pmpl-1*::mCherry. Recordings of cell lines without induction (top), 6–7 hrs after tetracycline addition (middle), and 28 hrs after tetracycline addition (bottom) are shown.
(PDF)

**S12 Fig. Tetracycline-induced toxicity in stable cell lines. (A)** Live-cell images of Control cells (CMV-driven eGFP; tetON::*pmpl-1*::mCherry) and Experimental cells (CMV-driven *peel-1*::eGFP; tetON::*pmpl-1*::mCherry). Cell lines are shown without tetracycline (−tet) and 24 hrs after tetracycline addition (+tet (24 hrs)). Insets in −tet conditions show leaky expression of PMPL-1 in both cell lines (LUT adjusted within inset). Images show successful tetracycline-inducible expression of PMPL-1::mCherry and efficient tetracycline-induced killing in experimental cells but not control cells. Arrows and inset in experimental cells +tet (24 hrs) show examples of swollen cells. Exposure time in the green channel is different between cell lines. Scale bar = 40 μm. **(B)** Time course of toxicity after addition of tetracycline to experimental cells. Noticeable cell swelling is seen after 6 hrs (arrows). Some acute swelling may also be visible at 4 hrs after addition of tetracycline. Scale bar = 20 μm.
(PDF)

**S13 Fig. Uninduced Experimental cells are permeable to monovalent cations.** Current–voltage plots of Experimental cells without tetracycline in different ionic conditions to test permeabilities of the indicated ions. Permeable ions have

greater inward currents (Na$^+$, K$^+$, Cs$^+$; closed symbols) compared to impermeable ions (NMDG$^+$, Ca$^{2+}$, Cl$^-$; open symbols) at negative voltages. Mean with SEM is shown. These data are summarized in Fig 5E. Underlying data are available in S2 Data.

(PDF)

**S14 Fig.  Predicted PEEL-1 pentameric structure.** AlphaFold2 prediction of the PEEL-1 pentameric structure is shown. Two angles are shown: **(A–B)** top view, showing the predicted extracellular face of the complex, and **(C–D)** side view, in the plane of the lipid bilayer. **(A and C)** Ribbon diagram with the amphipathic helix colored in blue creating the lining of a pore-like region. **(B)** Surface representation of electrostatic predictions (red = negative charge, blue = positive charge). An uninterrupted hole can be seen through the structure, with a ring of negative charge from five D109 residues. **(D)** Surface representation of hydrophobicity (yellow = hydrophobic, cyan = hydrophilic). PDB file available in S3 Data. **(E)** Cytotoxicity of mutants which eliminate the predicted ring of negative charge at the top of the complex via a D109A mutation. Plot shows mean with SD. Statistics performed using multiple unpaired *t*-tests with Holm-Šídák test. All tested comparisons are shown. Underlying data for (E) are available in S2 Data.

(PDF)

**S15 Fig.  PEEL-1 and PMPL-1 purification and liposomes. (A)** SDS–PAGE gels of indicated constructs after purification (P; Coomassie Blue stain and anti-MBP western blot) and after incorporation into liposomes (L; anti-MBP western blots). Liposomes were collected after flotation through a density gradient before use in synthetic bilayer experiments. The top fraction contains liposomes (L) and the bottom fraction (B) contains unincorporated protein. Monomers (m), oligomers or aggregates (o), and truncation products of PEEL-1 (t) are indicated. The bottom bands seen in both MBP-tagged PEEL-1 and PMPL-1 purifications are soluble proteins (s) that did not get incorporated into liposomes, likely endogenous MBP (43 kDa) or truncated MBP-tagged protein. **(B)** PEEL-1 is toxic to bacteria and toxicity requires the amphipathic helix. *E. coli* C41(DE3) cells with constructs encoding IPTG-inducible expression of either MBP::PEEL-1 (left) or MBP::PEEL-1(−65aa) (right). Images of cultures are taken after shaking at 18 °C overnight, with or without 0.5 mM IPTG. Cultures appear lysed when full-length PEEL-1 is expressed but not PEEL-1(−65aa) which lacks the PEEL-1 AH. Western blot (anti-MBP) of corresponding whole-cell lysates are shown (bottom), confirming higher expression of MBP::PEEL-1(−65aa). This experiment was repeated four times and yielded similar results. Original uncropped blots available in S1 Raw images.

(PDF)

**S16 Fig.  Purified PEEL-1 and PMPL-1 conduct ions through planar lipid bilayers.** Conductance traces through artificial planar lipid bilayers are shown. **(A)** Bilayer alone without addition of liposomes, **(B)** bilayer with PEEL-1 liposomes added (transient spikes indicate successful liposome fusions), **(C)** bilayer with PMPL-1 liposomes added, and **(D–E)** two independent experiments of bilayers with PEEL-1 and PMPL-1 liposomes added. An all-point histogram is shown for each trace (right, 2 pS bin width, normalized based on probability density). Channel activity after addition of PEEL-1 and PMPL-1 was observed in 8 independent experiments. A voltage of −180 mV was applied to the bilayer for liposome fusions in all experiments. After observing channel activity, a voltage of +180 mV was applied since channel activity was more stable at positive voltages. Bottom two panels show traces at +180 mV. Scale bar = 5 s. Underlying data are available in S2 Data.

(PDF)

**S1 Table.  *C. elegans* strains used in this study.**

(PDF)

**S2 Table.  Constructs used in this study.**

(PDF)

**S1 Movie. A single HEK293T cell co-expressing PEEL-1::eGFP and PMPL-1::mCherry.** Movie starts at 18.5 hrs after transfection. One frame = 5 min. Scale bar = 10 µm. Selected frames of this movie are shown in Fig 3C.
(MP4)

**S1 Data. Excel spreadsheet containing underlying numerical data for Figs 1B,1C,2A,2B,2E,3B,4C–4F,5A–5C,5E, and 5G.**
(XLSX)

**S2 Data. Excel spreadsheet containing underlying numerical data for S1B,S8,S9,S10,S13,S14E, and S16A–16E Figs.**
(XLSX)

**S3 Data. PDB file of AlphaFold2 prediction of the PEEL-1 pentamer shown in S14 Fig.**
(PDB)

**S1 Raw images. Original, uncropped blots for Figs 5D and S15A–B.**
(PDF)

## Acknowledgments

We thank Amy Clippinger, Chau Vuong, Irini Topalidou, Michael Crawford, Tyler Couch, Emma Mackey, Teresa Swanson, Andrew Oberst, Adam Steinbrenner, Sharona Gordon, Bertil Hille, Jihong Bai, Alexey Merz, Suzanne Hoppins, Harmit Malik, and members of the Malik lab (Fred Hutch) for their technical help and input throughout this project. We thank Aaron Severson for the AFS216 strain and Jérôme Cattin-Ortolá for isolating *yak52*. Some strains were provided by the CGC, which is funded by NIH Office of Research Infrastructure Programs (P40 OD010440).

## Author contributions

**Conceptualization:** Lews Caro, Aguan D Wei, Christopher A Thomas, Galen Posch, Michael Ailion.

**Data curation:** Lews Caro, Aguan D Wei, Christopher A Thomas, Galen Posch, Michaela C Franzi, Sarah J Abell.

**Formal analysis:** Lews Caro, Aguan D Wei, Christopher A Thomas, Galen Posch.

**Funding acquisition:** Lews Caro, Andrew H Laszlo, Jens H Gundlach, Jan-Marino Ramirez, Michael Ailion.

**Investigation:** Lews Caro, Aguan D Wei, Christopher A Thomas, Galen Posch, Ahmet Uremis, Michaela C Franzi, Sarah J Abell, Michael Ailion.

**Methodology:** Lews Caro, Aguan D Wei, Christopher A Thomas, Galen Posch, Ahmet Uremis, Andrew H Laszlo, Jens H Gundlach, Michael Ailion.

**Project administration:** Lews Caro, Michael Ailion.

**Resources:** Lews Caro, Aguan D Wei, Christopher A Thomas, Galen Posch, Ahmet Uremis, Michaela C Franzi, Sarah J Abell, Andrew H Laszlo, Jens H Gundlach, Jan-Marino Ramirez, Michael Ailion.

**Software:** Lews Caro, Aguan D Wei, Christopher A Thomas, Galen Posch.

**Supervision:** Andrew H Laszlo, Jens H Gundlach, Jan-Marino Ramirez, Michael Ailion.

**Validation:** Lews Caro, Aguan D Wei, Christopher A Thomas, Galen Posch, Ahmet Uremis, Michaela C Franzi, Sarah J Abell.

**Visualization:** Lews Caro, Aguan D Wei, Christopher A Thomas.

**Writing – original draft:** Lews Caro, Aguan D Wei, Christopher A Thomas.

**Writing – review & editing:** Lews Caro, Aguan D Wei, Christopher A Thomas, Andrew H Laszlo, Michael Ailion.

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
