## [Editor Report · Decision Letter 0]

5 Dec 2024

Dear Dr Ailion, 

Thank you for submitting your revised manuscript from Review Commons entitled "Mechanism of an animal toxin-antidote system" for consideration as a Research Article by PLOS Biology. Firstly, please accept my sincere apologies for the long delay in getting back to you with feedback on your revision as we discussed it with an academic editor with the relevant expertise. 

Your manuscript has now been evaluated by the PLOS Biology editorial staff and I am writing to let you know that we would like to send your submission back to the original reviewers at Review Commons. After discussions with the Academic Editor, whilst we agree with Reviewer's #1 and #3 that the additional experiments requested by Reviewer #2 are outside of the scope (e.g. structure determination and mutagenesis), we would like to recruit an additional referee with expertise in electrophysiology and purified protein biochemistry to provide a review of the existing data in the manuscript. 

IMPORTANT

After discussions within the editorial team and given the reviewer reports, we would like to consider your manuscript as a 'Discovery Report' at the journal (see editorial guidelines here: https://journals.plos.org/plosbiology/s/what-we-publish#loc-discovery-report). Upon resubmission (see details below), I would be grateful if you could please tick 'Discovery Report' as the article type in the dropdown menu in the online submission form. 

Before we can send your manuscript to reviewers, we need you to complete your submission by providing the metadata that is required for full assessment. To this end, please login to Editorial Manager where you will find the paper in the 'Submissions Needing Revisions' folder on your homepage. Please click 'Revise Submission' from the Action Links and complete all additional questions in the submission questionnaire.

Once your full submission is complete, your paper will undergo a series of checks in preparation for peer review. After your manuscript has passed the checks it will be sent out for review. To provide the metadata for your submission, please Login to Editorial Manager (https://www.editorialmanager.com/pbiology) within two working days, i.e. by Dec 07 2024 11:59PM.

Kind regards,

Richard

Richard Hodge, PhD

rhodge@plos.org

PLOS

---

## [Decision Letter · Decision Letter 1]

22 Jan 2025

Dear Michael,

Thank you for your patience while we considered your revised manuscript from Review Commons entitled "Mechanism of an animal toxin-antidote system" for publication as a Discovery Report at PLOS Biology. Please accept my sincere apologies for the delays that you have experienced during the peer review process, in part due to the closure of the editorial office during the Christmas holidays. Your revised study has been evaluated by the PLOS Biology editors, the Academic Editor and the original reviewers. As you know, we also recruited an additional reviewer (Reviewer #4) to assess the electrophysiological aspects of the study. 

As you will see, the previous reviewers at Review Commons are now satisfied with the revised version and recommend acceptance. In addition, the additional reviewer with expertise in planar lipid bilayer electrophysiology considers the data and conclusions to be sufficient. While we do appreciate the very positive comments of the reviewers, after discussing them with the Academic Editor, who has a background in biochemistry, we agreed that the reviewers missed some essential controls and should be added to the study to confirm the conclusions. The Academic Editor's comments are pasted below (labelled 'Comments from the Academic Editor') and they would be required in order for us to consider the manuscript for publication. This includes providing additional control experiments and reporting details to ensure that the purified PEEL-1 and PMPL-1 proteins are fully characterized and that the currents are recorded using physiologically relevant conditions to avoid artifacts. 

Given the extent of revision needed, we cannot make a decision about publication until we have seen the revised manuscript and your response to the reviewers' comments. Your revised manuscript is likely to be sent for further evaluation by all or a subset of the reviewers.

**IMPORTANT - SUBMITTING YOUR REVISION**

*Re-submission Checklist*

*Published Peer Review*

*PLOS Data Policy*

*Blot and Gel Data Policy*

Kind regards,

Richard

Richard Hodge, PhD

rhodge@plos.org

REVIEWS:

Reviewer #1 (Bruce A Hay, signs review): I appreciate the author comments and the changes to the discussion. 

My one minor comment in rereading the manuscript is in the beginning of the results. The authors say "Endogenous, sperm-delivered PEEL-1 is also

suppressed by pmpl-1 mutations (Fig. 1C), but not via a paternal-effect"

I think they mean "Killing by endogenous..."

Reviewer #2: I dont have further comments on the manuscript, which is appropriate to be published as a short discovery article.

Reviewer #3: The authors have addressed all concerns. I have no further issues.

Reviewer #4: I read the discussion and the revised manuscript. Unfortunately I m not an expert in the biological part so I will not comment on that. Concerning the electrophysiology I consider the data and the conclusion sufficient. Obviously a detailed study can be made but this would be study by its own. 

The cellular work shows pore forming activity and reconstitution in planar lipid bilayer a distribution of discrete conductances. As the overall study sounds very interesting I suggest to publish as it is. 

COMMENTS FROM THE ACADEMIC EDITOR

After carefully going over the manuscript again, I still have some concerns about the bilayer experiments. I think the authors should address these in their manuscript by providing appropriate control experiments and a better characterization of their data to provide stronger evidence that the observed currents in planar bilayers have equivalent properties to the currents measured in HEK-293 cells.

In light of the positive remarks of all four reviewers I regard the chance that the paper will be accepted as very high. I also do not think that it should be sent for re-review and that it could even be published without the bilayer data in case these turn out to be inconclusive.

The questions I have concerning protein expression, purification, reconstitution and bilayer recordings are as follows:

Protein expression and purification:

It was not confirmed that E. coli would be an appropriate overexpression system for both proteins, particularly since eukaryotic membrane proteins are only in exceptional cases functionally overexpressed in prokaryotic cells.

I also noticed that the authors extract their proteins after dounce-homogenization, which I do not regard as efficient in disrupting bacterial cells, followed by extraction in buffer containing 2% of the detergent Octyl-Glucoside. I thus wonder which fraction of the totally overexpressed proteins is captured in that way.

I also wonder why the authors record the absorbance at 260 and not 280 nm to estimate protein yields.

It would be helpful, if the authors would characterize their purified protein by size exclusion chromatography to estimate the oligomeric organization and dispersity of the sample and separate the fraction of the protein with appropriate oligomeric state from aggregates and incompletely assembled populations.

I think that this information is very important for several reasons:

1. Octyl glucoside is a comparably harsh detergent that is compatible with few stable membrane proteins (typically of prokaryotic origin) but that leads to aggregation in other cases.

2. The protein is expressed as MBP fusion, which tends to increase the solubility of misfolded membrane proteins and which was not cleaved off after purification.

3. Size exclusion chromatography could provide evidence for the proposed pentameric organization.

Bilayer recordings:

The authors should provide better evidence that the currents measured in their bilayer experiments have similar selectivity and current-voltage relationships as macroscopic currents. So far, the currents were only recorded at strongly positive voltages (180 mV) and at very high salt concentration (600 mM NaCl) and I do not know why the authors have decided for such drastic conditions.

I also wonder why these currents show the observed strong fluctuations and why they did not stabilize over time, particularly since macroscopic currents in Hek cells did not show a pronounced time dependence.

It would also be interesting to know whether the authors have observed stable macroscopic currents in their experiments that would facilitate the recording of meaningful current voltage relationships. If the spikes observed in Fig. 16 B, C would indeed correspond to the fusion of single vesicles with the bilayer, I would expect a constant current increase due to the incorporation of an increasing number of channels also in the recordings displayed in Fig. 16 D, E.

Comparisons of current voltage relationships and recordings in asymmetric ionic conditions would help to clarify these issues.

I regard these control experiments as essential since planar bilayer experiments are prone to artifacts and unfolded proteins have previously been shown to induce single-channel like properties.

---

## [Editor Report · Decision Letter 2]

27 Mar 2025

Dear Michael,

Thank you for your patience while we considered your revised manuscript "Mechanism of an animal toxin-antidote system" for publication as a Discovery Report at PLOS Biology. This revised version of your manuscript has been evaluated by the PLOS Biology editors and the Academic Editor.

Based on our Academic Editor's assessment of your revision, I am pleased to say that we are likely to accept this manuscript for publication, provided you satisfactorily address the following data and other policy-related requests that I have provided below (A-G):

(A) Your manuscript is currently being considered as Discovery Report (https://journals.plos.org/plosbiology/s/what-we-publish#loc-discovery-report) at the journal. Given the comments from Reviewer #2 during the last round of review and that the additional controls for the planar lipid bilayer assays will not be included, we would like to keep the article type as a Discovery Report. Our shorter format articles have a maximum of 4 main figures, so I would be grateful if you could reduce the number of main figures by 1 at this stage (either by combining two of the main figures or moving one figure to the supplementary). 

(B) We routinely suggest changes to titles to ensure maximum accessibility for a broad, non-specialist readership. In this case, we would suggest a minor edit to the title, as follows. Please ensure you change both the manuscript file and the online submission system, as they need to match for final acceptance:

“PEEL-1 toxin kills cells by co-opting the PMPL-1 membrane protein to create a novel cation channel”

(C) You may be aware of the PLOS Data Policy, which requires that all data be made available without restriction: http://journals.plos.org/plosbiology/s/data-availability. For more information, please also see this editorial: http://dx.doi.org/10.1371/journal.pbio.1001797

-Supplementary files (e.g., excel). Please ensure that all data files are uploaded as 'Supporting Information' and are invariably referred to (in the manuscript, figure legends, and the Description field when uploading your files) using the following format verbatim: S1 Data, S2 Data, etc. Multiple panels of a single or even several figures can be included as multiple sheets in one excel file that is saved using exactly the following convention: S1_Data.xlsx (using an underscore).

-Deposition in a publicly available repository. Please also provide the accession code or a reviewer link so that we may view your data before publication. 

Figure 1B-C, 2A-B, 2E, 3B, 4C-F, 5A-C, 5E, 5G, S1B, S8, S9, S10, S13, S14E, S16

(D) Please also ensure that each of the relevant figure legends in your manuscript include information on *WHERE THE UNDERLYING DATA CAN BE FOUND*, and ensure your supplemental data file/s has a legend.

(E) We require the original, uncropped and minimally adjusted images supporting all blot and gel results reported in the following Figures:

Figure 5D, S15A-B

We will require these files before a manuscript can be accepted so please prepare and upload them now. Please carefully read our guidelines for how to prepare and upload this data: https://journals.plos.org/plosbiology/s/figures#loc-blot-and-gel-reporting-requirements

(F) Per journal policy, if you have generated any custom code during the course of this investigation, please make it available without restrictions. Please ensure that the code is sufficiently well documented and reusable, and that your Data Statement in the Editorial Manager submission system accurately describes where your code can be found. 

(G) Please ensure that your Data Statement in the submission system accurately describes where your data can be found and is in final format, as it will be published as written there. 

We expect to receive your revised manuscript within two weeks. 

*Published Peer Review History*

*Press*

Best regards,

Richard

Richard Hodge, PhD

rhodge@plos.org

PLOS

---

## [Editor Report · Decision Letter 3]

28 Apr 2025

Dear Michael,

On behalf of my colleagues and the Academic Editor, Raimund Dutzler, I am pleased to say that we can accept your manuscript for publication, provided you address any remaining formatting and reporting issues. These will be detailed in an email you should receive within 2-3 business days from our colleagues in the journal operations team; no action is required from you until then. Please note that we will not be able to formally accept your manuscript and schedule it for publication until you have completed any requested changes.

PRESS

Best regards, 

Richard

Richard Hodge, PhD

rhodge@plos.org

PLOS
